# Use of Amplified Lewy Body Dementia Fibrils and Autoradiography to Characterize Binding of Radioligand Tg-1-90B to Alpha-Synuclein Fibrils in Postmortem Brain Tissue

**DOI:** 10.3390/cells14181477

**Published:** 2025-09-22

**Authors:** Jennifer Y. O’Shea, Dhruva D. Dhavale, Helen Hwang, Zachary Smith, Thomas J. A. Graham, Robert H. Mach, Paul T. Kotzbauer

**Affiliations:** 1Department of Neurology, Hope Center for Neurological Disorders, Washington University School of Medicine, St. Louis, MO 63110, USA; jenniferoshea@wustl.edu (J.Y.O.); dhavaled@wustl.edu (D.D.D.);; 2Department of Radiology, Perelman School of Medicine, University of Pennsylvania, Philadelphia, PA 19104, USArmach@pennmedicine.upenn.edu (R.H.M.)

**Keywords:** Tg-1-90B, alpha-synuclein fibrils, Parkinson’s disease, radioligand binding, amplified LBD fibrils, Lewy body dementia, fibril structure, autoradiography

## Abstract

Parkinson’s disease (PD) and Lewy Body Dementia (LBD) are defined by accumulation of alpha-synuclein (Asyn) fibrils within Lewy bodies (LBs) and Lewy neurites (LNs). The development of a Positron Emission Tomography (PET) tracer for quantifying Asyn fibrils would improve diagnostic accuracy and provide a biomarker for disease progression. We previously described radioligand [^3^H]Tg-1-90B, which binds to in vitro Asyn fibrils (PDB 2N0A) via interactions with residues Y39, S42 and K44. Here, we performed molecular docking studies with Tg1-90B and PD/LBD Asyn fibrils (PDB 8A9L), which predicts interactions with residues Y39 and K43 in a structurally distinct binding site. In radioligand binding assays, Tg-1-90B has moderate to high affinity (K_d_ 17.5 nM) for amplified LBD fibrils (PDB 8FPT), whose protofilament fold is highly similar to PD/LBD fibrils (PBD 8A9L). Autoradiography confirmed binding of [^3^H]Tg-1-90B to LBs in PD brain tissue. However, Tg-1-90B also binds to amyloid-beta fibrils in Alzheimer’s disease (AD) tissue, indicating insufficient selectivity for Asyn fibrils. These results indicate that Tg-1-90B binds to Asyn fibrils in PD tissue but needs further structural optimization. Binding assays with amplified LBD fibrils and autoradiography with postmortem PD tissue can guide further development of Asyn fibril PET ligands for PD/LBD.

## 1. Introduction

Parkinson’s disease (PD) and Lewy Body Dementia (LBD) are neurodegenerative disorders defined by the abnormal aggregation and accumulation of alpha-synuclein (Asyn) protein [1,2,3]. Asyn is a small 140 amino acid presynaptic protein, abundantly expressed throughout the nervous system and highly soluble in normal brain tissue [1,4,5,6]. In PD, Asyn misfolds and forms insoluble fibrils stabilized by beta sheet structure. The accumulation of Asyn fibrils, in cytoplasmic inclusions called Lewy bodies (LB), and in neuritic inclusions called Lewy neurites (LN), is the defining neuropathological feature of PD and LBD at autopsy [7,8].

An Asyn imaging agent that reliably detects pathologic Asyn accumulation in PD would improve the accuracy of diagnosis [9]. The accuracy of clinical diagnosis of idiopathic PD ranges from 76% to 92% [10,11]. Developing an imaging technique to quantify fibrillar Asyn in vivo would also provide a biomarker for disease progression [12]. However, developing an Asyn imaging agent presents several challenges. First, an Asyn imaging ligand must have high affinity (lower than 10 nM) for pathologic Asyn fibrils, since the concentration of Asyn fibrils in PD is lower than that of other fibrillar protein species, such as amyloid-beta (Aβ) and tau. Second, an Asyn imaging agent must have more than 10 fold selectivity for Asyn fibrils over Aβ and tau fibrils, since many cases of PD have widespread accumulation of Aβ and rare cases have widespread accumulation of tau [9,13,14,15,16].

Current Asyn ligand development efforts mostly rely on the use of binding assays with in vitro prepared Asyn preformed fibrils (PFFs) to screen small molecules. Recent advances in cryo-EM and SSNMR have demonstrated that different Asyn fibril conformers are produced by incubating human recombinant monomeric Asyn protein under different conditions to induce fibril formation, including differences in pH, buffers and salts [17,18,19]. Recently, Yang et al. reported a 2.2 Å structure (PDB 8A9L) of Asyn fibrils extracted from PD and LBD postmortem tissue, determined by single particle cryo-EM analysis [20]. The structure of these ex vivo PD and LBD fibrils is significantly different from the structures of all in vitro PFF preparations reported to date. Given the limited quantities of Asyn fibrils that can be isolated from the brain and with the goal of recapitulating Asyn fibril formation that occurs in vivo, our laboratory recently produced an amplified Asyn fibril preparation starting from seeds extracted from LBD postmortem brain tissue, called amplified LBD fibrils [21]. Using SSNMR and cryo-EM, we observed that the fold of amplified LBD fibrils (PDB 8FPT) is highly similar to the fold of the ex vivo PD/LBD fibrils (PBD 8A9L).

We previously identified a set of chalcone and styrene-based analogs that bind to Asyn fibrils using Thioflavin-T competition assays [15]. Molecular docking studies with the styrene analog Tg-1-90B identified residues Y39-S42-T44 (Site 2) as the putative binding site on the in vitro Asyn fibrils having a “Greek-key β-sheet topology” as determined by solid state NMR [15,22]. Since the structure of these fibrils (PBD 2NOA) differs from the structure of ex vivo fibrils from PD and LBD, we investigated whether this ligand can also bind to postmortem PD brain tissue. We first performed molecular docking studies with different Asyn fibril structures to understand the putative binding site of Tg-1-90B. We used radioligand saturation binding assays with amplified LBD fibrils to characterize the binding affinity of [^3^H]Tg-1-90B to Asyn fibrils. Then, we performed nuclear emulsion autoradiography with LBD, Alzheimer’s disease (AD) and control postmortem tissue. We observed that [^3^H]Tg-1-90B binds to LBs present in tissue and that binding to LBs can be displaced by excess unlabeled ligand. Therefore, the use of amplified LBD fibrils in conjunction with autoradiography provided confirmation that Tg-1-90B has a binding site in PD postmortem tissue, making this radioligand a valuable scaffold to identify new ligands.

## 2. Materials and Methods

### 2.1. General

All reagents were purchased from Sigma-Aldrich (St. Louis, MO, USA) unless otherwise indicated.

### 2.2. Ethical Statement and Use of Postmortem Brain Tissue

The Movement Disorders Brain Bank, Washington University, St. Louis, MO, USA, provided clinically and neuropathologically well-characterized postmortem frozen brain tissue. Written informed consent to perform a brain autopsy was obtained from all participants. After death, the immediate next-of-kin were contacted and confirmed consent for brain removal and retention of brain tissue for research purposes. This brain donation program is independent of this study. The use of postmortem brain tissue is not categorized as human studies research according to Washington University in St. Louis Human Research Protection Office (HRPO) and IRB approval was not applicable. The method and characterization of LBD amplified fibrils has been described in Dhavale et al. [21]. The PD case for autoradiography were selected based on a clinical diagnosis of PD plus dementia, Braak LB stage 5–6 pathology, and the absence of significant Aβ or tau pathology determined by immunohistochemistry. Disease-control case was selected based on the absence of Asyn, Aβ and tau pathology. AD case was selected with clinical diagnosis of dementia, Braak stage C for Aβ pathology, and Braak stage V–VI for tau pathology.

### 2.3. Molecular Docking of Tg-1-90B

Compound Tg-1-90B was drawn with Chem3D v25.0 (PerkinElmer, Waltham, MA, USA). Molecular docking studies were performed via AutoDock Vina [23] (Scripps Research, San Diego, CA, USA) and viewed with Pymol (pymol.org, accessed on 18 September 2025). Full-length recombinant in vitro Asyn fibril (PDB 2N0A), ex vivo LBD Asyn fibrils (PDB 8A9L), and amplified LBD fibrils (PDB 8FPT) were obtained from RCSB protein data bank (https://www.rcsb.org/, accessed on 18 September 2025) as a target protein for blind docking experiments. Water molecules and nonpolar hydrogens were removed from compounds and protein structures. Docking to in vitro fibrils (PDB 2N0A) was performed as previously described [15]. We increased structures 8A9L and 8FPT to 5 fibril units using RELION v5.0 [24] and kept only the beta-sheet regions containing amino acids residues (aa) 31–100. For 8A9L, a grid box with a dimension of 95 × 50 × 95 Å and center x = 100, y = 100, and z = 100 was applied. For 8FPT, a grid box with a dimension of 100 × 100 × 100 Å and center x = −33, y = 0, and z = −11 was applied. Energy range was set to 7 and exhaustiveness was set to 8 (default values).

### 2.4. Preparation of IV_Tris_ Fibrils (Recombinant PFF), Amplified LBD Fibrils and Tissue Insoluble Fraction (IF) Homogenates

IV_Tris_ fibrils were assembled in vitro by spontaneous nucleation in 20 mM Tris-HCl, pH 8.0, 100 mM NaCl buffer in a total volume of 1500 µL at 2 mg/mL monomer concentration. The incubation was carried out at 37 °C with continuous shaking at 1000 rpm in a Thermomixer (Eppendorf, Framingham, MA, USA) for 72 h. Amplified LBD fibrils were prepared as described in Dhavale et al. and case LBD1 amplified fibrils [21] were used in this study. PD, disease-control and AD insoluble fraction (IF) tissue homogenate was prepared by sequential homogenization of dissected tissue gray matter in a series of detergents followed by centrifugation to obtain insoluble fraction. The detailed procedure to make IV_Tris_ fibrils, amplified LBD fibrils and tissue insoluble fraction homogenates is described in Dhavale et al. [21].

### 2.5. Preparation of Amyloid-Beta (1-42) (Aβ) and Tau Fibrils

Synthetic amyloid-beta 1–42 (Aβ) and recombinant tau fibrils were prepared according to methods described in Bagchi et al. [16].

### 2.6. In Vitro Saturation Binding Assays of [^3^H]Tg-1-90B

A fixed concentration of binding substrates IV_Tris_ fibrils (250 nM), amplified LBD fibrils (50 nM), Aβ fibrils (150 nM), tau fibrils (150 nM), PD, AD or disease control homogenate (10 µg/mL) were incubated for 2 h at 37 °C with increasing concentrations of [^3^H]Tg-1-90B in a reaction volume of 150 µL in a 96-well plate. Assay buffer was 30 mM Tris-HCl pH 7.4, 0.1% BSA. [^3^H]Tg-1-90B (hot) to unlabeled (cold) Tg-1-90B ratio of 1:10-30 was used to generate the concentration curve. In parallel, a similar reaction was set up with the addition of 2.5 µM unlabeled (cold) Tg-1-90B to determine nonspecific binding. After incubation, bound and free radioligands were separated by vacuum filtration through 1.0 μm glass fiber filters in 96-well filter plates (Catalog MSFBN6B50, MilliporeSigma, St. Louis, MO, USA), followed by three 150 µL washes with cold assay buffer. Filters containing the bound ligand were mixed with 150 µL of Optiphase Supermix scintillation cocktail (PerkinElmer, Waltham, MA, USA) and counted following overnight incubation. All data points were performed in triplicate. The dissociation constant (K_d_) and the maximal number of binding sites (B_max_) values were determined by fitting the data to the equation Y = B_max_ × X/(X + K_d_) by non-linear regression using Graphpad Prism software (version 7.0).

### 2.7. Nuclear Emulsion Autoradiography with Postmortem Brain Tissue Sections

We performed nuclear emulsion autoradiography with PD, AD and control postmortem brain tissue sections to confirm target engagement of Tg-1-90B with pathologic Asyn, Aβ, tau and any other additional target. Brain samples were sectioned to 16 µm and mounted onto glass slides. The brain sections were dried for 15 min and stored at −80 °C. For each experiment, adjacent frozen brain sections were fixed in 4% paraformaldehyde (PFA) at room temperature for 15 min. To quench lipofuscin autofluorescence, slides were counterstained with 20× TrueBlack for 1 min. LBs and LNs were immunostained with (2.5 µg/mL in assay) P-syn/81A (epitope phosphorserine 129, Catalog 825701, Biolegend, San Diego, CA, USA) and Alexa Fluor 594 prior to radioligand incubation. Similarly, AD sections were stained with (2.3 µg/mL in assay) Alexa-568 conjugated to mHJ3.4 (a kind gift from John Cirrito laboratory) to detect Aβ_1–42_ in amyloid plaques. Antibody PHF1 (epitopes S396 and S404, a kind gift from the late Peter Davies) and Alexa Flour 488 conjugated goat anti-mouse were used for immunostaining neurofibrillary tangles, which are primarily composted of tau. The control tissue was immunostained with P-syn/81A to confirm absence of Asyn pathology, as that was the primary target of Tg-1-90B. The [^3^H]Tg-1-90B unblocked condition was diluted in 30 mM Tris-HCl, pH 7.4 to bring the final concentration of radioligand to 40 nM. For blocked conditions, unlabeled Tg-1-90B was added at a competing concentration of 500 nM to evaluate the potential displacement of [^3^H]Tg-1-90B binding.

After radioligand incubation for 2 h at 37 °C, the slides were removed from the respective radioactive solutions and briefly incubated in a series of washes to remove unbound radiotracer. Wash solutions and incubation times were as follows: 30 mM Tris-HCl, pH 7.4 for 1 min, 50% ethanol/30 mM Tris-HCl, pH 7.4 for 1 min, 30% ethanol/30 mM Tris-HCl, pH 7.4 for 1 min, 30 mM Tris-HCl, pH 7.4 for 1 min, and lastly, a quick rinse in MQH_2_O for 1 s. To obtain high-resolution autoradiographic information, the slides were coated with Ilford K5D nuclear emulsion (Catalog 10580, HARMAN technology Ltd, Cheshire, WA, USA). Each slide was vertically dipped in emulsion for 10 s, removed and allowed to drain for 10 s, and dipped again for 10 s. After letting the emulsion drain for 10 s, the excess emulsion was scraped off the back of the slide and then stored in an Adorama Paper Safe, Light-Tight Protection box that gets vacuumed sealed and kept away from sources of radioactivity for a duration of 4 days. After the exposure time, the slides were removed in a darkroom and developed as follows: 200 mL of 1 + 4 Ilford Phenisol Developer (Catalog 10147, HARMAN technology Ltd, Cheshire, WA, USA) for 4 min, 200 mL of 1% acetic acid for 2 min, 200 mL of 1 + 4 Ilford Hypam Fixer (HARMAN technology Ltd, Cheshire, WA, USA) for 4 min, and 200 mL of MQH_2_O for 4 min (2×). The slides were allowed to completely air dry in the fume hood before getting coverslipped with Fluoromount-G Mounting Medium Catalog 0100-01, SouthernBiotech, Birmingham, AL, USA). Photomicrographs of tritium activated silver grains and fluorescent antibody staining were captured on a Nikon Eclipse TE2000-U Inverted Fluorescence Microscope (Nikon, Tokyo, Japan) microscope using the brightfield/monochrome and fluorescence channel, respectively.

### 2.8. Quantification of Nuclear Emulsion Autoradiography in Postmortem Brain Tissue

Photomicrographs were collected of LBs to determine binding of [^3^H]Tg-1-90B in postmortem brain tissue and the quantification of radioligand activated silver grains was performed in Image J v1.54 (Fiji). The antibody-stained LB from the fluorescent imaging channel was first overlayed on top of the corresponding silver grain image from the brightfield channel. The ‘freehand selection’ of the LB provided a boundary used to define and measure the area of silver grain colocalization. The overlay was then removed to measure the silver grains independently and to avoid fluorescent signal interference. The ‘Analyze’ function in ImageJ v1.54 was used to measure the mean gray value (or the sum of gray values of all the pixels in the selection divided by the number of pixels) of the selected area of silver grains. In total, 18 LBs and 12 LBs were quantified for the [^3^H]Tg-1-90B radioligand only and radioligand plus excess unlabeled ligand slides, respectively. Data were analyzed using GraphPad Prism software, version 10 (GraphPad Software, Inc., La Jolla, CA, USA). Due to unequal group sizes and nonnormal distribution of data points, nonparametric Mann–Whitney U tests were utilized to compare groups.

## 3. Results

### 3.1. Structural Comparison of Ligand Binding Sites Among Asyn Fibril Conformers

We compared the structural configuration of residues Y39, S42 and T44 in three Asyn fibril conformers: in vitro Asyn fibrils (PDB 2N0A [22]); ex vivo LBD Asyn fibrils (PDB 8A9L [20]) extracted from PD/LBD postmortem brain tissue; and amplified LBD fibrils (PDB 8FPT [21]) (Figure 1). In our previous manuscript, we predicted binding interactions of Tg-1-90B with these three residues (Figure 1A) [15]. These residues form a small molecule binding pocket within the Greek key fold of the in vitro Asyn fibrils but have a different configuration in ex vivo fibrils (PDB 8A9L) (Figure 1B) and amplified LBD fibrils (PBD 8FPT) (Figure 1C), where resides S42 and T44 are oriented away from Y39 and appear unlikely to be accessible for interaction with small molecule ligands. Therefore, we performed molecular docking simulations with these Asyn structures to understand the predicted binding sites of Tg-1-90B on these fibrils.

### 3.2. Molecular Docking of Tg-1-90B

We performed a series of molecular docking simulations using Tg-1-90B to dock with in vitro Asyn fibrils (PDB 2N0A), ex vivo LBD Asyn fibrils (PDB 8A9L) and amplified LBD fibrils (PDB 8FPT). For in vitro Asyn fibrils with a Greek-fold (PDB 2N0A), the docking simulation identified a binding pocket at residues Y39-T44, comprising interactions with the Y39, S42 and T44 side chains, consistent with our previous observation (Figure 2A,B). When Tg-1-90B was docked to ex vivo Asyn fibrils (PDB 8A9L) (Figure 2C,D) and amplified LBD fibrils (PDB 8FPT) (Figure 2E,F), a different binding site pocked was identified at residues Y39-K43, comprising interactions with the Y39 and K43 sides chains. Thus, molecular docking confirmed our qualitative assessment that binding of Tg-1-90B with different Asyn fibril conformers is likely to involve different binding site structures and distinct interactions with amino acid side chains. To further investigate binding interactions of Tg-1-90B with different Asyn fibril conformers, we performed binding studies with an in vitro Asyn fibril preparation and with amplified LBD Asyn fibrils (8FPT), whose protofilament fold is highly similar to the fold of ex vivo LBD Asyn fibrils (8A9L).

### 3.3. [^3^H]Tg-1-90B Binds to IV_Tris_ Asyn and Amplified LBD Fibrils

To evaluate binding properties for in vitro PFFs, we utilized IV_Tris_ fibrils prepared by incubation of recombinant Asyn monomer at 37 °C with continuous shaking. Using negative stain EM, these fibrils appear distinct from LBD amplified fibrils, with helical two protofilament morphology [21]. The protofilament fold of IV_Tris_ fibrils is highly similar to that of the Greek-key fold of Asyn fibrils 2N0A used in previous docking studies (manuscript in preparation). We also performed binding assays with amplified LBD fibrils (8FPT), whose protofilament fold is highly similar to the fold of fibrils extracted from PD postmortem brain tissue (8A9L), to further assess the binding properties of Tg-1-90B. For the IV_Tris_ fibrils, we observed one site saturable specific binding with an average K_d_ of 40.1 ± 1.9 nM with an average B_max_ of 22.3 ± 0.1 pmol/nmol (Figure 3A). For amplified LBD fibrils, we observed an average K_d_ of 17.5 ± 4.3 nM with an average B_max_ of 20.6 ± 2.5, indicating higher binding affinity for amplified LBD fibrils compared to in vitro Asyn fibrils (Figure 3B).

We further performed [^3^H]Tg-1-90B saturation binding assays with Aβ and tau fibrils to assess selectivity of Tg-1-90b. Aβ fibrils were made from a synthetic Aβ_1–42_ peptide, while recombinantly prepared tau monomer was used to generate tau fibrils. [^3^H]Tg-1-90B bound to both synthetic Aβ fibrils and recombinant tau fibrils. We observed one site saturable specific binding with an average K_d_ of 31 ± 1.7 nM and an average B_max_ of 8.6 ± 2.4 pmol/nmol for Aβ fibrils (Appendix A). Similarly, we observed one site saturable specific binding with an average K_d_ of 48.7 ± 13 nM and an average B_max_ of 7.5 ± 0.6 pmol/nmol for tau fibrils (Appendix A). These results indicate that [^3^H]Tg-1-90B has only modest selectivity for amplified LBD fibrils over Aβ and tau fibrils.

### 3.4. [^3^H]Tg-1-90B Binding Assays with Tissue Homogenates

We also performed binding assays with insoluble fraction (IF) tissue homogenates prepared from PD (Figure 4A), disease-control (Figure 4B) and Alzheimer’s disease (AD) postmortem brain tissue (Figure 4C). We observed low specific to nonspecific binding for both PD and disease-control tissue homogenates. Due to low levels of specific binding of [^3^H]Tg-1-90B in PD tissue, we could not accurately determine the binding affinity in PD tissue. In control IF, total binding was marginally higher than nonspecific binding, with no indication of high affinity specific binding in control IF, which would indicate off-target binding if present. We obtained an average K_d_ value of 25.7 ± 3.7 nM in AD tissue IF, and the specific to nonspecific ratio for the AD tissue homogenate assay was much higher compared to the PD tissue homogenate assay, which is likely explained by the substantially higher concentration of Aβ and tau fibrils in AD tissue compared to the concentration of Asyn fibrils in PD tissue.

### 3.5. [^3^H]Tg-1-90B Binds to Lewy Bodies in Postmortem Brain Tissue

To further validate and confirm the results of the radioligand binding assays, we used autoradiography to determine whether Tg-1-90B binds to Asyn fibrils accumulating in PD brain tissue, the in vivo target for an Asyn PET tracer for PD. As an alternative to phosphorimaging autoradiography, we used nuclear emulsion autoradiography to evaluate binding of [^3^H]Tg-1-90B in PD postmortem brain tissue (Figure 5). Silver halide crystals from nuclear emulsion exposure to sections incubated with [^3^H]Tg-1-90B consistently colocalized with anti-Asyn antibody (P-syn/81a) immunostaining of LBs (Figure 5A–C). In contrast, sections blocked with excess unlabeled Tg-1-90B did not show accumulation of silver halide crystals over LBs, confirming that [^3^H]Tg-1-90B binding to LBs is displaceable (Figure 5D–F). We were not able to detect consistent labeling of LNs, likely due to the smaller thread-like size of these structures, which likely results in signal that is below the sensitivity of detecting radioligand binding by nuclear emulsion. Quantification of [^3^H]Tg-1-90B autoradiography signal for the [^3^H]Tg-1-90B incubated (unblocked) and [^3^H]Tg-1-90B plus unlabeled Tg-1-90B sections (blocked) confirmed specific binding to LBs, based on significant and substantial displacement by excess unlabeled Tg-1-90B consistently (Figure 5G). We did not observe any apparent off-target specific binding in the autoradiography studies with PD tissue, with the acknowledgement that this technique has low sensitivity for assessing this question, which is addressed more quantitively in radioligand binding assays with fibrils and tissue homogenates from PD, AD and control autopsy cases.

### 3.6. [^3^H]Tg-1-90B Autoradiography with AD and Control Postmortem Brain Tissue

To further characterize the selectivity and specificity of [^3^H]Tg-1-90B, we performed nuclear emulsion autoradiography with AD and control postmortem brain tissue sections. We chose AD tissue as it contains both pathologic Aβ plaques and neurofibrillary tangles, which are primarily composed of tau. Control tissue was chosen based on the absence of Asyn, Aβ and tau pathology. Results showed that [^3^H]Tg-1-90B bound to Aβ plaques, as confirmed by co-localization by anti-Aβ antibody (mHJ3.4) immunostaining of Aβ plaques in AD tissue. The binding of [^3^H]Tg-1-90B to plaques was displaceable using excess unlabeled Tg-1-90B (Appendix A). In contrast, we observed no specific binding of [^3^H]Tg-1-90B to neurofibrillary tangles in AD tissue (Appendix A). Furthermore, we did not observe any specific pattern/distribution of [^3^H]Tg-1-90B accumulation in control tissue, indicating that [^3^H]Tg-1-90B does not have significant off-target binding other than the Aβ plaques (Appendix A).

## 4. Discussion

In this study, we utilized molecular docking, radioligand binding assays and autoradiography to characterize the binding of styrene-based radioligand [^3^H]Tg-1-90B to IV_Tris_ fibrils, amplified LBD fibrils and postmortem brain tissue. Molecular docking of Tg-1-90B predicted a binding site for in vitro Asyn fibrils (PDB 2N0A) that involves structurally distinct interactions compared to the binding site predicted for ex vivo (PDB 8A9L) and amplified LBD fibrils (PDB 8FPT). Consistent with this prediction of structurally distinct binding sites, we observed different binding affinities for in vitro Asyn fibrils and amplified LBD fibrils in radioligand binding assays, with [^3^H]Tg-1-90B displaying moderate affinity (K_d_ 40 nM) towards in vitro Asyn fibrils and higher affinity (K_d_ 17.5 nM for amplified LBD fibrils. In PD tissue homogenate binding assays, [^3^H]Tg-1-90B did not generate a sufficiently high specific-to-nonspecific binding ratio needed to characterize binding properties, but nuclear emulsion autoradiography confirmed that [^3^H]Tg-1-90B binds specifically to Lewy bodies (LBs) in PD postmortem brain tissue, with displacement by unlabeled ligand. [^3^H]Tg-1-90B showed moderate affinity binding to synthetic Aβ and recombinant tau fibrils, and displayed similar binding affinity in binding assays with postmortem AD tissue homogenate. Interestingly, [^3^H]Tg-1-90B bound to Aβ plaques but not to tau tangles in AD postmortem brain tissue. Taken together, the results indicate that Tg-1-90 does not have sufficiently high affinity for Asyn fibrils or selectivity vs. other amyloid fibrils to justify further development as an Asyn imaging agent. Nevertheless Tg-1-90B can be utilized as a scaffold for developing an Asyn PET ligand, with the potential to improve its affinity and selectivity for PD Asyn fibrils with further structural optimization.

The development of Asyn PET has been challenging due to the limited quantities of Asyn fibrils present in the PD brain tissue and the occurrence of other pathologic amyloid aggregates such as amyloid-beta and tau fibrils in a subset of PD autopsy cases. For instance, in our autopsy study analyzing postmortem brain tissue, we found widespread Aβ pathology in 60% of cases while widespread tau pathology was found in 3% of cases [25], underscoring the need for PET tracers with high affinity, ideally a K_d_ less than 10 nM and close to 1 nM, plus high selectivity. To date, several Asyn PET tracers have been published, including phenothiazine derivatives (SIL series) [16], benzothiazoles, styrene derivatives, BF 227 [26], Anle253b derivates [27,28], ACI-12589 [29], MPBB3 derivatives [30], F0502B [14] and M503 [31], but it is not yet clear whether any can achieve the needed affinity, selectivity and in vivo binding properties required for imaging Asyn fibril accumulation in PET studies of individuals with PD/LBD [32]. Therefore, further effort is needed to optimize existing leads and identify new leads in developing an Asyn PET tracer for PD PET imaging.

Pre-clinical identification of Asyn PET ligands often relies initially on binding assays using in vitro Asyn fibrils (PFFs), which are generated by spontaneous nucleation of the Asyn monomer in vitro. Advances in cryo-EM and SSNMR have elucidated the structures of many different PPF polymorphs, all of which have a substantially different fold compared to the structure of the ex vivo LBD fibrils. In contrast, the fold of amplified LBD fibrils is highly similar to the fold of ex vivo PD/LBD fibrils and may provide a useful screening tool to bridge the gap between PFFs and postmortem tissue studies that include binding assays and autoradiography.

In this study, we were not able to accurately measure the binding properties of [^3^H]Tg-1-90B in postmortem PD brain tissue homogenates, likely due to the relatively low concentration of Asyn fibrils, producing a low ratio of specific to nonspecific binding. We did not observe any significant high affinity specific binding in disease-control tissue homogenates, which would indicate off-target binding if present. The amount of nonspecific binding observed with in vitro binding assays utilizing tissue homogenates may not predict nonspecific binding in vivo, where it is likely to be driven by in vivo pharmacokinetic properties of the ligand at low concentrations. We observed higher specific to nonspecific binding ratio in AD homogenate, with moderate affinity binding that corresponds to the moderate affinity binding we observed for in vitro Aβ and tau fibrils. The higher specific binding in AD tissue is likely explained by the fact that the concentration of binding substrate in AD tissue, namely Aβ fibrils, is 10–20 fold higher than that of Asyn fibrils in LBD/PD brain tissue for a cortical region like the middle frontal gyrus [25].

Given the low concentration of Asyn fibrils in PD brain tissue and the moderate affinity of Tg-1-90B, we decided to pursue nuclear emulsion autoradiography instead of phosphorimager autoradiography in PD postmortem brain tissue. The primary goal in the nuclear emulsion autoradiography with PD tissue was to determine target engagement for Asyn fibrils in LBs. Nuclear emulsion autoradiography allows for visualization of target engagement utilizing substantially higher resolution imaging compared to autoradiography using a phosphorimager or beta imager, with sensitivity that is typically equal to or greater than that of phophorimager autoradiography. Phosphorimagers and beta imagers enable faster quantification of autoradiography results, but the precision with which signal can be correlated with immunostaining for LBs is limited by the lower resolution. By developing the silver halide crystals activated by bound radioligand, we observed specific target engagement of Tg-1-90B with LBs based on fluorescence immunostaining of LBs with an anti-Asyn antibody. Importantly, binding of [^3^H]Tg-1-90B to LBs was substantially displaced by excess cold ligand, indicating specific binding of the radioligand in the context of radioligand binding properties, where binding is determined to be specific if it is displaced by excess cold ligand and nonspecific if it is not displaced by excess cold ligand [33]. We were not able to visualize binding of Tg-1-90B to LNs, likely related to the small size of these structures which produces signal that is below the sensitivity of the nuclear emulsion technique. Therefore, the binding of Tg-1-90B to LNs is not characterized. In autoradiography studies with AD tissue, [^3^H]Tg-1-90B specifically detected Aβ plaques but not neurofibrillary tangles, which contain tau fibrils. This contrasted with the binding observed to recombinant tau fibrils in saturation binding assays. Previous studies indicate that the structure of in vitro assembled recombinant tau fibrils is significantly different from the structure of tau fibrils isolated from AD postmortem tissue [34,35]. We did not observe any other off-target binding of Tg-1-90B in control tissue. Taken together, these results further highlight the need for using fibril preparations that closely resemble the pathologic aggregates that exist in brain tissue, to obtain accurate measures of binding affinity for a ligand. Structure studies with single particle cryo-EM and SSNMR can also be utilized in future studies to further characterize ligand–fibril interactions.

Using nuclear emulsion, we were able to visualize Tg-1-90B binding at a molecular level, showing its interaction with LBs and Aβ plaques. It is worth noting that the production of nuclear emulsion for this autoradiography approach requires a specialized production process that limits its availability for widespread use. However, this technique is ideally suited to confirm target engagement of radioligands, when the targets can be identified histologically with techniques such as fluorescence immunostaining. Binding affinity is an important consideration for assessing off-target binding and can be accurately assessed in radioligand binding assays where specific and nonspecific binding are measured at different ligand concentrations. In contrast, the lower throughput nature of autoradiography experiments makes it difficult to obtain binding affinity measurements.

The combination of radioligand binding assays with amplified LBD fibrils and autoradiography studies with postmortem PD tissue sections allowed us to overcome challenges with radioligand binding assays using tissue homogenates and to characterize the binding properties of Tg-1-90B for Asyn fibrils in LBD postmortem brain tissue. Our results indicate that Tg-1-90B can serve as a scaffold for further ligand development to obtain the desired binding affinity and selectivity for Asyn fibrils in PD/LBD tissue. Future studies can utilize competitive binding assays with [^3^H]Tg-1-90B and amplified LBD fibrils to identify analogs of Tg-1-90B with improved binding affinity, as well as for compounds from other structural scaffolds that may interact at the same binding site. These studies can be complimented with selectivity assessments using competitive binding assays with AD tissue homogenates. Optimized ligands can then be further validated in autoradiography studies with PD, AD and control tissue before evaluating brain uptake in biodistribution studies and in vivo binding in animal models of Asyn fibril accumulation. In vivo binding studies with an animal model for Asyn fibril accumulation can include microPET imaging as well as ex vivo autoradiography analysis, where fibril levels could be determined by immunostaining in parallel with quantitative autoradiography.

## 5. Conclusions

Our results demonstrate that Tg-1-90B binds to amplified LBD Asyn fibrils and specifically binds to Lewy bodies in PD brain tissue. Our results indicate that Tg-1-90B does not have sufficiently high affinity and selectivity for in vivo imaging of Asyn fibrils. However, the structure of Tg-1-90b provides a scaffold for further ligand development. Binding assays with amplified LBD fibrils and autoradiography with PD, AD and control tissue can be utilized to identify analogs with improved binding properties.

## 6. Patents

Authors specified below have a patent application pending titled “Tissue-Seeded Fibrils and Methods of Making and Using Same”. Patent applicant: Washington University in St. Louis, Name of inventors: Paul T. Kotzbauer, Dhruva D. Dhavale, and Jennifer Y. O’Shea, Application number: 17/858817, Status of application: Pending. The patent application covers the process of generating amplified fibrils, its methods and composition.

## Figures and Tables

**Figure 1 cells-14-01477-f001:**
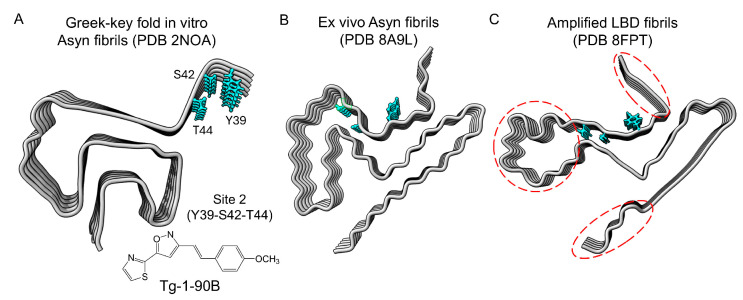
Qualitative structural comparison across different Asyn fibril conformations for residues which form the binding “Site 2” for Tg-1-90B on in vitro Asyn fibrils. The predicted binding site for radioligand Tg-1-90B containing residues Y39-S42-T44 (site 2, residues in blue) is displayed on (**A**) in vitro Asyn fibrils with a Greek-key fold (PDB 2N0A). These same residues and their side chains are displayed on (**B**) ex vivo Asyn fibrils extracted from postmortem brain tissue (PDB 8A9L) and (**C**) amplified LBD fibrils (PDB 8FPT). Dashed red circles on the structure of amplified LBD fibrils indicate regions not resolved by SSNMR.

**Figure 2 cells-14-01477-f002:**
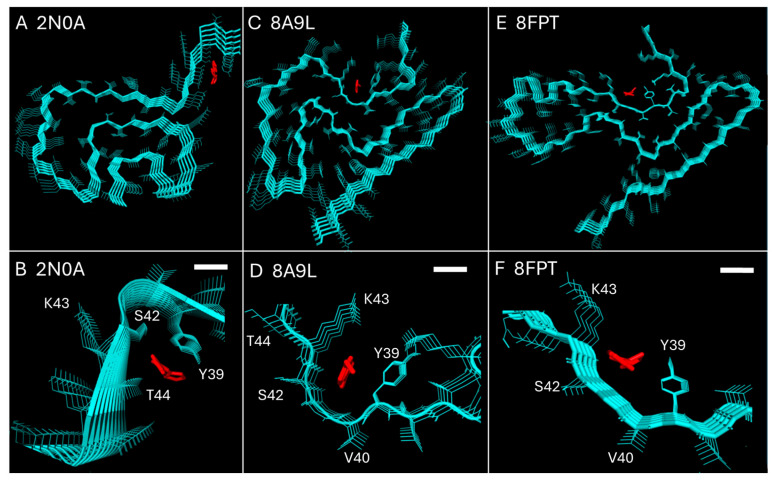
Tg-1-90B binding to Asyn fibrils as predicted by molecular docking. Molecular docking simulations were performed using Tg-1-90B and in vitro Asyn fibrils (PDB 2N0A), ex vivo LBD Asyn fibrils (PDB 8A9L) and amplified LBD fibrils (PDB 8FPT). Results suggest a different binding site pocket containing different residues when Tg-1-90B binds to recombinant fibrils (2N0A) (**A**,**B**) versus ex vivo LBD fibrils (PDB 8A9L) (**C**,**D**) and amplified LBD fibrils (PDB 8FPT) (**E**,**F**). The top panels (**A**,**C**,**E**) are of the beta sheets as viewed from the fibril axis while bottom panels (**B**,**D**,**F**) are zoomed in to the binding pocket. Ligand Tg-1-90B is in red and scale bars (3.8 Å) indicating the distance between carbons in consecutive amino acids on the protein’s main chain are shown in (**B**,**D**,**F**).

**Figure 3 cells-14-01477-f003:**
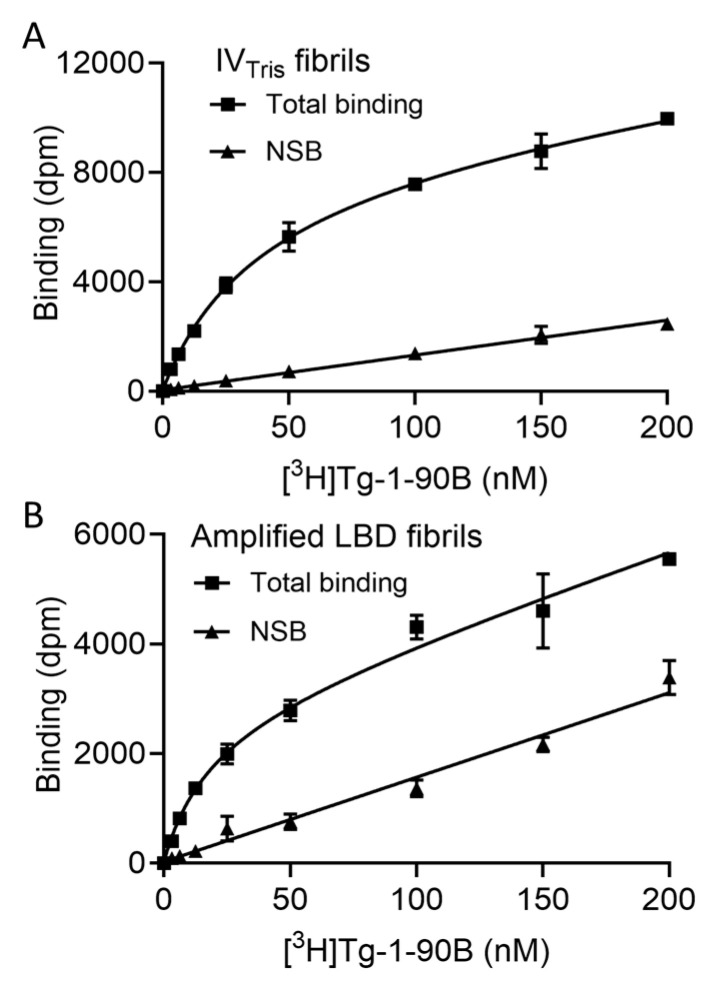
[^3^H]Tg-1-90B saturation binding assays with IV_Tris_ and amplified LBD fibrils. Saturation binding assays were performed as described and total, along with nonspecific binding (NSB), was analyzed by global fit non-linear regression analysis in GraphPad. (**A**) The average binding affinity (K_d_) of [^3^H]Tg-190B to IV_Tris_ fibrils was 40.1 ± 1.9 nM with an average B_max_ of 22.3 ± 0.1 pmol/nmol Asyn. (**B**) The average binding affinity (K_d_) of [^3^H]Tg-190B to amplified LBD fibrils was 17.5 ± 4.3 nM with an average B_max_ of 20.6 ± 2.5 pmol/nmol Asyn. Representative curves are shown and data points represent mean ± s.d. Similar results were obtained from *n* ≥ 2 independent assays.

**Figure 4 cells-14-01477-f004:**
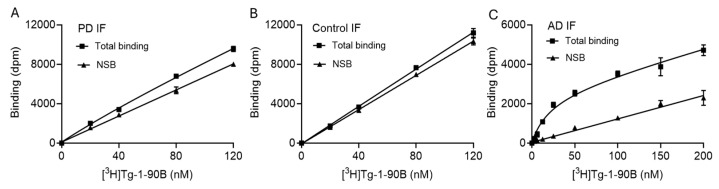
Binding of [^3^H]Tg-1-90B to insoluble fraction (IF) from PD, disease-control and AD tissue. [^3^H]Tg-1-90B saturation binding assays with PD (**A**), disease-control (**B**) and AD IF (**C**) were performed as described and total, along with nonspecific binding (NSB), was analyzed by global fit non-linear regression analysis in GraphPad. Due to low ratios of specific to nonspecific binding, the binding affinity of [^3^H]Tg-1-90B could not be accurately determined in PD or control tissue homogenate. We observed a binding affinity (K_d_) of 25.7 ± 3.7 nM with a B_max_ of 567 ± 28.3 pmol/g. wet wt. to AD tissue using the one site binding model with linear regression analysis. Data points represent mean ± s.d and *n* ≥ 2 independent assays.

**Figure 5 cells-14-01477-f005:**
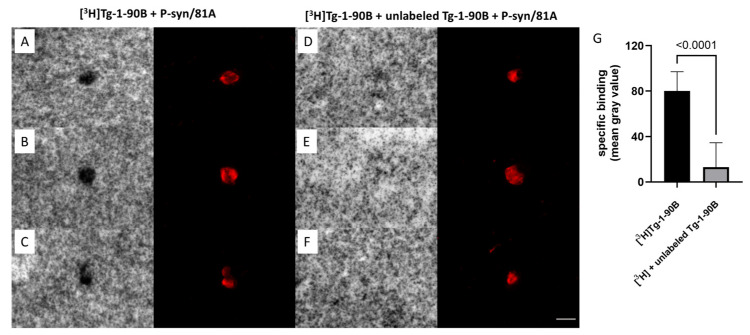
Nuclear emulsion autoradiography confirms binding of [^3^H]Tg-1-90B to Lewy bodies in human postmortem PD brain tissue. (**A**–**C**) Silver halide crystals in the nuclear emulsion are activated by [^3^H]Tg-1-90B, resulting in dense silver grain accumulation that colocalized on the LBs identified by immunoflourescence staining with anti-Asyn antibody P-syn/81A (red). (**D**–**F**) Addition of 500 nM of excess unlabeled (cold) Tg-1-90B displaced the radioligand [^3^H]Tg-1-90B, resulting in the absence of silver grain accumulation over LBs as indicated by P-syn/81A (epitope: Serine 129) immunostaining (red). Quantitative autoradiography performed on [^3^H]Tg-1-90B and [^3^H]Tg-1-90B plus unlabeled cold sections by measuring silver grain accumulation over LBs shows specificity of [^3^H]Tg-1-90B towards Asyn fibrils in PD tissue (**G**). In total, 18 LBs and 12 LBs were quantified for [^3^H]Tg-1-90B and [^3^H]Tg-1-90B plus unlabeled Tg-1-90B sections with [^3^H]Tg-1-90B exposed sections showing significant difference compared to [^3^H]Tg-1-90B plus unlabeled Tg-1-90B sections (*p* < 0.0001). No significant accumulations of silver grains were observed in association with Lewy neurites. Representative images are shown and similar results were obtained in two independent experiments. Scale bar in panel (**F**) = 10 µm.

## Data Availability

The original contributions presented in this study are included in the article/Appendix A. Further inquiries can be directed to the corresponding authors.

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
