# Peer review of "Use of Amplified Lewy Body Dementia Fibrils and Autoradiography to Characterize Binding of Radioligand Tg-1-90B to Alpha-Synuclein Fibrils in Postmortem Brain Tissue"

_cells, 2025, doi:10.3390/cells14181477_

Round 1
Reviewer 1 Report (Previous Reviewer 2)
Comments and Suggestions for Authors
In the article entitled “Radioligand Tg-1-90B binds to amplified Lewy body dementia fibrils and to Lewy bodies in postmortem tissue,” the authors use a radioligand that binds to Lewy body fibrils.
Although there were some changes to the manuscript, there are still some important points to address.
- Moderate affinity (Kd~17.5 nM for amplified LBD fibrils) and lack of the required high selectivity (>10×) against Aβ and tau fibrils, which limits its immediate value as a PET tracer.
- There is a low specific binding ratio in PD tissue homogenates, making it impossible to accurately characterize affinity.
- Limited sensitivity of the nuclear emulsion to detect fine structures such as Lewy neurites, leaving a gap in ligand characterization.
- The study lacks a quantitative analysis of the correlation between fibril levels and binding signal in different brain regions.
Suggestions.
- Optimize the structure of Tg-1-90B to increase affinity (ideally Kd < 10 nM) and improve selectivity against Aβ and tau, using structure-activity relationship studies and analog screening.
- Incorporate competition assays with reference ligands and kinetic assessments to better understand ligand-fibril interaction.
- Complement autoradiography with more sensitive or higher resolution techniques to detect smaller fibrils (e.g., Lewy neurites).
4. Conduct regional tissue studies to correlate fibril density and signal, which could better guide optimization and validate utility as a biomarker.
Author Response
Please see the attachment.

Reviewer 2 Report (Previous Reviewer 1)
Comments and Suggestions for Authors
Review Cells-3842293
Radioligand Tg-1-90B binds to amplified Lewy body dementia fibrils and to Lewy bodies in postmortem tissue
Jennifer Y. O’Shea et al
This manuscript is a revision of original submission Cells-3705712. The authors have added additional experiments and modified their initial conclusions.
The additional data and discussion have modified the scope and conclusions of the manuscript so that the main assertion seems to be that probe Tg-1-90B may be utilized as a scaffold from which new ligands may be identified. My comments stem from this new assertion.
- The manuscript contains molecular docking results of Tg-1-90B to different isoforms of alpha synuclein fibrils, but the accompanying experiments do not relate to these docking simulations in any way, and also do not show a viable path toward using these results in further development involving Tg-1-90B. The putative sites highlighted in the simulations are not tested, and since the probe has been demonstrated to also bind to Abeta and tau fibrils, the binding to these additional proteins should be included if the authors intend to discuss Tg-1-90B binding at the molecular level.
- In the modified Discussion; “[3H]Tg-1-90B showed moderate affinity binding to synthetic Aβ and recombinant tau fibrils, which is also evident in binding assays with postmortem AD tissue homogenate.”
This has not been experimentally determined, as the authors note in the latter parts of their discussion:
“The higher specific binding in AD tissue is likely explained by the fact that the concentration of binding sub- strate in AD tissue, namely Aβ fibrils, is 10-20 fold higher than that of Asyn fibrils in LBD/PD brain tissue for a cortical region like middle frontal gyrus[29]”
It seems possible, however, to determine if this hypothesis is correct by performing analyses analogous to those in Figure 5, using anti Abeta and Tau antibodies instead of synuclein antibodies and see if colocalization may be observed in AD postmortem tissue samples.
The end result of addressing these two points would be to further strengthen the proposition that Tg-1-90B may be useful in further development toward a useful PET probe, and additionally, the authors might add some specific routes of inquiry in these further studies, based on their results.
Minor point; lane 289 of the revised manuscript; “nonspecific binging” -> binding?
Round 2
Reviewer 1 Report (Previous Reviewer 2)
Comments and Suggestions for Authors
Thanks to the authors for their replies.
The revised version presents significant improvements, including the addition of autoradiography experiments in Alzheimer’s disease and control brain tissue, which enhance the overall evaluation of the radioligand’s specificity. The authors critically address the moderate-to-high affinity of Tg-1-90B for α-synuclein fibrils and its off-target binding to Aβ and tau, providing a transparent assessment of the ligand's limitations. The study employs a rigorous multi-method approach, including in vitro binding assays, autoradiography, and immunofluorescence with well-characterized postmortem tissue samples. The manuscript is clearly written, the figures are improved, and the topic is highly relevant for the development of α-synuclein PET tracers.
Reviewer 2 Report (Previous Reviewer 1)
Comments and Suggestions for Authors
I have read the resubmitted manuscript and find that the authors have addressed all of my comments satisfactorily.
This manuscript is a resubmission of an earlier submission. The following is a list of the peer review reports and author responses from that submission.
Round 1
Reviewer 1 Report
Comments and Suggestions for Authors
This manuscript describes the design and characterization of a small molecule radioligand, 3HTg-1-90B, that targets deposits of alpha-synuclein in Lewy bodies. The probe is designed for use in determining deposits within brain tissue samples, using Positron Emission Tomography. Probe Tg-1-90B is a styrene-based analog molecule identified from assays using in vitro formed synuclein fibrils, and a candidate site of binding was estimated from docking studies.
In the present study, stipulating that the molecular characteristics between synuclein fibrils that are formed in vivo are distinct from fibrils isolated from actual patients or fibrils amplified in vitro from tissue-derived fibril seeds, the authors probe if 3HTg-1-90B is capable of specifically binding to and identifying Lewy Body deposits that contain alpha synuclein.
My concerns regarding this submission all involve the proposed “specificity” of this probe toward alpha synuclein fibrils, as much of the data and information given by the authors seem to indicate that probe specificity might be problematic for this probe. Comparision of the fibril structures of in vitro(PDB 2N0A), brain tissue-derived (8A9L) and amplified (8FPT) do not immediately suggest a binding site within the latter two structures for a molecule identified by docking to the former structure; the probe shows non-negligible amounts of non-specific binding, and what specific binding that could be detected were for samples isolated from Alzheimer’s disease patients, not Parkinson’s disease patients, and in Figure 4, there were no negative control experiments that could be performed in light of the results in Figure 3C, that probe binding to Alzheimer’s disease tissue samples. The chase experiments in Figure 4, where an excess of cold Tg-1-90B is used to suppress signals from 3HTg-1-90B, is not in the strict test a test of specificity, since the two molecules are identical, and therefore dilution of the radioactive signal is expected.
I believe that the authors should attempt to prove or disprove that their newly developed probe is in fact specific toward alpha-synuclein, rather than a broad specificity probe toward beta fibril structure in general. Experiments should utilize negative control experiments, for example comparisons between syn fibril and tau fibril, determination of the relative amounts of synuclein relative to the other protein components in samples (Figures 3 and 4), and binding inhibition experiments using antibodies for competition assays, since antibodies are more targetable.
In its present form, I am unable to agree with the manuscript’s conclusion that “Tg-1-90B binds to both IVTris and amplified LBD Asyn fibrils and specifically binds to Lewy bodies in PD brain tissue.”
Minor points; Many spelling mistakes are visible, especially in the former part of the manuscript.
- Line 96-97 “20 mm” ->20 mM
- Line 125, 132, lack of spaces between value and unit
- Line 142 “for10”
- Line 176 “resides”
- Legend to Figure 2, “ [3H]Tg-190B”, two places
Reviewer 2 Report
Comments and Suggestions for Authors
In the article entitled “Radioligand Tg-1-90B binds to amplified Lewy body dementia fibrils and Lewy bodies in postmortem tissue” the authors developed a radioligand targeting alpha-synuclein to improve the diagnosis of diseases such as Parkinson's and dementias with Lewy bodies.
There are several points that need to be addressed.
- Limited ligand affinity in tissue: Although there is binding to LBs in sections, binding in PD tissue homogenates could not be adequately quantified due to the low specific/non-specific ratio.
- Potential non-specific binding: Low signals were observed even in control tissue, which could indicate interaction with undesired targets.
- In vivo selectivity is not addressed: The study lacks functional data in ex vivo animal or brain models assessing whether the ligand penetrates blood-brain barrier or has real diagnostic utility.
- No correlation with clinical imaging studies: There are no correlations with real PET studies or evaluation in preclinical in vivo models.
- Little discussion of technical limitations: Although the low amount of fibrils in tissue is briefly mentioned, a more in-depth discussion of the sensitivity of the techniques used (e.g., nuclear emulsion vs. standard autoradiography) would be desirable.
- Regarding Figure 1(F1): Key molecular distances between residues are not provided.
- Partial representation(F1): Only 3 residues are shown without context of the three-dimensional environment of the complete ligand fold.
- No docking models are included(F1): The figure would be more informative if it showed the docked Tg-1-90B in all three structures to visualize the impact of conformational change in docking.
- Regarding Figure 2(F2): The variability of Bmax in IVTris is high (±9.9), which may reflect inconsistencies in fibril preparation or ligand accessibility.
- No visible negative control(F2): No non-specific binding curves are shown in the figure; although they are mentioned in the text, their visual inclusion would facilitate interpretation.
- Limited range of Kd(F2): Although Kd < 20 nM is acceptable, it is still above the ideal threshold (<10 nM) for a PET radiotracer candidate.
- Regarding Figure 3(F3): The radioligand does not show good specific discrimination in PD homogenate, which limits its direct use without fractionation or enrichment.
- Apparent nonspecific binding in control(F3): The presence of signal suggests possible binding to structures other than Asyn.
- Poor fit to model in AD(F3): Although a Kd was estimated, the fit was not optimal, which questions the reliability of the value.
- Lack of normalization or quantitative control (F3): Total protein or fibril load values are not presented, which complicates the comparative interpretation.
- Regarding Figure 4(F4): The study acknowledges that no reliable signal was detected in these structures, possibly due to their small size or low fibril density.
- Limited number of structures quantified(F4): Only 18 LBs (unblocked condition) and 12 (blocked condition) were analyzed. Although sufficient to observe differences, a larger sample would be appreciated.
- No control or AD(F4) tissue was evaluated: Binding specificity to LBs compared to other aggregates or in non-pathological brains is not visualized in this figure.
19. Detailed statistical analysis is not reported(F4): Although the difference is mentioned, neither the p-value nor the type of statistical test used is indicated.